# Altered Immune Response to the Epstein–Barr Virus as a Prerequisite for Multiple Sclerosis

**DOI:** 10.3390/cells11172757

**Published:** 2022-09-04

**Authors:** Fabienne Läderach, Christian Münz

**Affiliations:** Viral Immunobiology, Institute of Experimental Immunology, University of Zurich, 8057 Zurich, Switzerland

**Keywords:** Epstein–Barr virus, multiple sclerosis, infectious mononucleosis, HLA-DR2, molecular mimicry, tertiary lymphoid structures, CNS inflammation

## Abstract

Strong epidemiologic evidence links Epstein–Barr virus (EBV) infection and its altered immune control to multiple sclerosis (MS) development. Clinical MS onset occurs years after primary EBV infection and the mechanisms linking them remain largely unclear. This review summarizes the epidemiological evidence for this association and how the EBV specific immune control is altered in MS patients. The two main possibilities of mechanisms for this association are further discussed. Firstly, immune responses that are induced during a symptomatic primary EBV infection, namely infectious mononucleosis, might be amplified during the following years to finally cause central nervous system (CNS) inflammation and demyelination. Secondly, genetic predisposition and environmental factors might not allow for an efficient immune control of the EBV-infected B cells that might drive autoimmune T cell stimulation or CNS inflammation. These two main hypotheses for explaining the association of the EBV with MS would implicate opposite therapeutic interventions, namely either dampening CNS inflammatory EBV-reactive immune responses or strengthening them to eliminate the autoimmunity stimulating EBV-infected B cell compartment. Nevertheless, recent findings suggest that EBV is an important puzzle piece in the pathogenesis of MS, and understanding its contribution could open new treatment possibilities for this autoimmune disease.

## 1. Introduction on EBV in Autoimmune Diseases

The Epstein–Barr virus (EBV) is a common human γ-herpesvirus, which persistently infects more than 95% of the adult human population, making it one of the most successful pathogens [1]. The EBV persists in memory B cells during latent infection and produces infectious virus particles upon the plasma cell differentiation of B cells and presumably after epithelial cell infection [2]. Once the virus has gained access to the memory B cell pool, it can persist without the expression of any viral proteins, the so-called latency 0. However, EBV encoded small noncoding RNAs (EBERs) are still expressed and can be detected by in situ hybridization [3]. Identified as the first human tumor virus, EBV is primarily associated with several lymphomas, carcinomas, as well as smooth muscle tumors [4]. However, its associated tumorigenesis that accounts for around 1–2% of all tumors worldwide is still rare considering EBV’s abundant distribution in the population [5]. One reason for this is the immune system’s ability to keep the premalignant infectious state under control [6]. This immune control of EBV infection needs to be in a perfect balance, as the hyperactivation of T cell compartments, often observed during inadequately controlled primary EBV infection, can lead to immune pathologies, such as infectious mononucleosis (IM) or hemophagocytic lymphohistiocytosis (HLH) [7,8]. In rare cases, such immune dysregulation can manifest itself in autoimmune diseases. Indeed, a dysregulation of viral persistence in the EBV infected memory B cell pool, as well as defective humoral and cellular immune responses against EBV have been reported in multiple autoimmune diseases [9,10,11,12].

As a matter of fact, this is the case in one of the most common autoimmune diseases, rheumatoid arthritis (RA). Peripheral blood of RA patients displays an abnormally high frequency of circulating EBV-infected B cells [12]. Additionally, the EBV infection of cultivated lymphocytes from RA patients yielded a higher amount of B-cell-transformed lymphoblastoid cell lines than those from healthy donor lymphocytes. Since an analysis of the general T cell function revealed no difference between the two groups, a more specific defect in the EBV specific suppressor T cell function has been proposed [13]. Similar observations have been described in the autoimmune disease systemic lupus erythematosus (SLE). Characterized by flares of disease activity and phases of remission, the disease course fits well with EBV’s ability to occasionally switch from a latent infection to its lytic cycle [14]. Patients suffering from SLE have an abnormally high EBV viral burden in their peripheral blood compared to healthy controls [15]. Furthermore, SLE patients have a tenfold increased frequency of EBV infected B cells [16]. One study showed that whole blood stimulated with EBV yielded a decreased amount of EBV specific cytotoxic CD8^+^ T cells producing IFN-γ in SLE patients, showing that this inability of SLE patients to control the EBV lytic cycle can be attributed to a reduced EBV specific T cell reactivity [15].

In autoantibody-driven diseases such as RA and SLE, elevated EBV viral loads might originate from lytic viral replication that arises secondary to autoimmune B cell differentiation into plasma cells [17,18], but might nevertheless contribute to the inflammatory environment of these autoimmune diseases. In contrast, EBV’s mechanistic contribution to multiple sclerosis (MS) remains largely unknown and contrary to RA and SLE, EBV dysregulation seems to be a prerequisite rather than a consequence of the autoimmune disease. Strong epidemiologic evidence has linked EBV infection to the development of MS. Essentially, nearly complete seropositivity (99.5–100%) has been reported in MS patients, compared to the healthy adult population with around 95% seroconversion [19,20,21]. While there are reports of EBV seronegative MS patients, those cases are extremely rare. It is even considered that the majority of EBV seronegative adults may be misdiagnosed as uninfected, since serological tests against multiple antigens are recommended to accurately define EBV status, and in the majority of EBV seronegative adults but not seronegative children, viral loads and EBV specific T cell responses could previously be detected [22,23]. Furthermore, individuals with a history of IM have a 3.2 higher risk of developing MS in comparison with individuals who acquired the virus asymptomatically [24]. A recent landmark study showed that primary EBV infection precedes clinical MS by several years and only after seroconversion were biomarkers for neuroaxonal degeneration increased [21]. However, the majority of people infected with EBV will never develop MS. This indicates, that EBV infection is a prerequisite which allows the development of the autoimmune disease in genetically susceptible individuals that are exposed to additional environmental factors. In the following, the main hypotheses how EBV, in certain predisposed individuals, might mechanistically contribute to MS pathology is discussed in more detail.

## 2. EBV Infection Precedes MS by Several Years

The low concordance rate for MS development in monozygotic twins of 20–30% argues for environmental influences on the development of this autoimmune disease [25]. The respective environmental risk factors seem to be geographically distributed with elevated risk for MS development in high northern or southern latitudes [26]. One reason for this could be the limited sunlight exposure and hence lower vitamin D levels in countries with a high latitude [27]. Indeed, vitamin D intake was shown to decrease the risk for MS development [28]. This geographically associated risk seems to be established early in life [26,29,30]. Those who migrate before the age of 12–15 years acquire to some extent the MS risk of the new geographical environment, while those who migrate later in life often retain the MS prevalence of their geographical origin, even so MS manifests clinically most often in early adulthood. Interestingly, the acquisition of a primary EBV infection differs in a similar geographic fashion. While primary EBV infection is nearly uniformly experienced prior to the age of 2 at low latitudes, such as equatorial Africa, and the immune pathology of IM is virtually unknown, in high latitudes EBV is acquired by around one third of the population later and often in the second decade of life with up to half of them experiencing IM [8,31]. There are indications that the low vitamin D level could be partially responsible for this geographical distribution of IM occurrence. Serum levels of 25-hydroxyvitamin D were shown to positively correlate with the ability of regulatory T cells to suppress T cell proliferation. This ability plays an important part during primary EBV infection, in controlling the EBV specific T cell response, thereby preventing immunopathologies as seen in IM [32,33]. Therefore, the elevated MS risk after IM of 3.2 and of 7 in individuals carrying the main genetic risk factor for MS, the HLA-DRB1*1501 MHC class II allele [24,34], might in part explain the geographical distribution of MS. 

The time interval between primary EBV infection and clinical manifestation of MS was investigated in longitudinal patient cohorts [19,21]. In these cohorts, EBV seroconversion occurred an estimated 7.5 years prior to the clinical onset of MS [21]. This sign of primary EBV infection increased the risk for MS development 32-fold compared to seronegative controls. All but one of the 955 retrospectively analyzed MS patients seroconverted prior to MS onset. This delay between primary EBV infection and clinical MS onset might be due to a preclinical phase of central nervous system (CNS) damage. Indeed, in this longitudinal study, serum levels of neurofilament light chain (NfL), a biomarker that has been associated with neuroaxonal damage [35], increased shortly after EBV infection. Therefore, it is an attractive hypothesis that in genetically predisposed individuals, delayed primary EBV infection with IM sets a pathogenic process in motion that after more than five years can lead to clinical symptoms of MS (Figure 1). 

Such an EBV driven mechanism is also suggested by the second parameter of EBV infection that is associated with MS risk, namely elevated antibody responses against EBV nuclear antigen 1 (EBNA1) [36,37,38,39]. In particular, elevated antibodies to the amino acid sequence 385–420 in HLA-DRB1*1501 positive individuals increase the MS risk 24-fold [40]. EBNA1 specific antibodies increase at least five years prior to clinical MS onset [37,38,39]. Similar to NfL levels, they also correlate with MS disease activity and likelihood to progress from clinically isolated syndrome (CIS) to MS [41,42,43,44]. Interestingly in the recent longitudinal study that identified EBV infection as a prerequisite for MS development [21], antibody responses to 200 virus species prior and post MS onset were assessed and only EBV and specifically EBNA1 specific antibody responses were significantly elevated in MS patients. Thus, altered EBV specific immune responses, often emerging upon delayed primary infection in genetically MS-susceptible individuals, might predispose for MS and develop several years before clinical MS onset. 

## 3. Molecular Mimicry Development during Infectious Mononucleosis

One possibility by which elevated EBNA1 specific antibody titers could directly influence MS development would be if they cross-reacted with myelin antigens, i.e., molecular mimicry would occur between EBNA1 and MS autoantigens. Indeed, EBNA1 and, in particular, its domain (aa385–420) that elicits antibodies associated with the biggest increase in MS risk [40] or the neighboring aa431–440 sequence provides homologies to several CNS autoantigens, including GlialCAM and anoctamin [45,46]. The presence of GlialCAM and EBNA1 cross-reactive antibodies in the cerebrospinal fluid of MS patients as part of disease-associated oligoclonal bands (OCB) could also be demonstrated [46] (Figure 1). EBNA1 specific antibodies as part of the OCBs had also been previously described [47,48,49]. The preconditioning of mice by vaccination with the EBNA1 peptide that is recognized by GlialCAM cross-reactive antibodies facilitated experimental autoimmune encephalomyelitis (EAE) induction in one animal model of MS [46]. This raises the question by which mechanism antibodies are primed when they cross-react between EBNA1 and autoantigens. One attractive scenario is that these autoreactivities could be initiated during IM, as myelin oligodendrocyte glycoprotein (MOG)-specific antibodies have been detected in a minority of IM patients [50]. It seems unlikely that EBV infection directly rescues autoreactive B cells that can produce such autoantibodies during IM, since EBV was not found to preferentially infect autoreactive B cells nor to enrich these in the memory B cell compartment [51]. However, the possibility still remains that high viral loads during IM broaden EBNA1 specific B cell responses to epitopes with molecular mimicry to CNS autoantigens. 

These cross-reactive antibody responses, especially those targeting anoctamin, which are associated with HLA-DRB1*1501 expression [45], possibly indicate an underlying viral and/or autoantigen-specific HLA-DRB1*1501-restricted CD4^+^ T cell response. Indeed, EBNA1 specific CD4^+^ T cell responses are also elevated in MS patients [11,52]. A subset of the EBNA1 specific CD4^+^ T cell clones cross-reacted with a peptide mixture derived from CNS autoantigens [52]. They also were capable of producing IL-2, IFN-γ and TNF-α, while only EBNA1 reactive T cell clones preferentially produced IFN-γ, TNF-α, MIP-1α and MIP-1ß. The cross-reactivity of CD4^+^ T cells of MS patients between the autoantigen RASGRP2 and the lytic EBV antigens BHRF1 and BPLF1 was also described [53]. Furthermore, MOG-specific CD4^+^ T cells were also stimulated with a peptide from the EBV DNA polymerase BALF5, another lytic EBV antigen [54]. Finally, CD4^+^ T cells that had been primed in mice with reconstituted human immune system components (humanized mice) and isolated on the basis of their HLA-DRB1*1501-restricted recognition of EBV transformed B cells were able to recognize myelin basic protein (MBP) [55]. Interestingly, the respective T cell clones could also recognize allogeneic EBV transformed B cell lines, at least by cytokine production, suggesting a broadly cross-reactive potential of these T cells. Under which circumstances these cross-reactive CD4^+^ T cells are primed remains unclear but their frequent HLA-DRB1*1501 restriction would explain additive MS risk due to HLA-DRB1*1501 expression and EBV infection. 

## 4. Defective Immune Control of EBV in MS Patients

The elevated EBNA1 antibody levels and EBNA1 specific CD4^+^ T cell frequencies in MS patients [11,36,52,56] might also reflect poorly controlled EBV infection in these individuals, even if most studies have not found significantly elevated blood viral loads [52,57]. MS patients have also a significantly higher frequency of IFN-γ-secreting EBV specific CD8^+^ T cells than healthy controls or patients with other neurological diseases [58]. In addition, CD8^+^ T cells specific for lytic EBV antigens were found to increase during active disease and decrease during phases of remission in MS [59]. Interestingly, in EBV infected humanized mice, EBNA1 specific IgM responses and activated CD8^+^ T cell frequencies correlated with EBV viral loads [55]. Similarly, total CD8^+^ T cell numbers in blood correlated with whole blood EBV viral loads in IM patients [60]. Therefore, elevated EBNA1 specific antibodies and T cell responses could reflect a poorly controlled compartment of EBV infected B cells in MS patients that is, however, compartmentalized and does not extensively communicate with the blood where EBV viral loads are not increased in MS patients. 

Such poorly controlled EBV infected B cells could migrate to the brain, where their presence has been detected by some but not other groups [61,62,63,64,65]. Along these lines, EBV infected B cells could be adapted for increased brain homing in mice [66]. This adaptation led to upregulation of EBNA1, secreted phosphoprotein 1/osteopontin (SPP1/OPN), neuron navigator 3 (NAV3), CXCR4 and germinal-center-associated signaling and motility protein (GCSAM). Blocking osteopontin could significantly compromise brain homing of EBV infected B cells in this experimental model. Despite the controversy on EBV infected B cell homing to the CNS of MS patients, B cell infiltration has been detected in MS brains and especially in progressive MS disease, and B cell infiltration and follicle formation in the CNS correlate with the severity of disease progression [67,68]. A distinct activated Tbet^+^ CXCR3^+^ memory B cell population has been reported to be preferentially enriched in the CNS compartments of MS patients, a phenotype found to be sustained by chronic EBV infection in mice [69,70]. To what frequency and extent EBV infected B cells are involved in the formation of these structures remains unclear but they could be involved in stimulating autoimmunity (Figure 1). Indeed, increased brain homing autoreactive CD4^+^ T cell activation upon autologous memory B cell stimulation has been observed in MS patients [71].

A decreased immune control of EBV could be particularly pronounced in the context of the main genetic risk factor for MS, HLA-DRB1*1501, and both IM as well as elevated EBNA1 specific antibody titers have been shown to interact with HLA-DRB1*1501 for additive MS risk [34,36,72,73]. Accordingly, humanized mice reconstituted with HLA-DRB1*1501 positive donor stem cells showed increased CD8^+^ T cell expansion and activation alongside elevated viral loads compared to HLA-DRB1*1501 negative or HLA-DRB1*0401 positive engrafted animals [55]. Thus, EBV specific immune control based on HLA-DRB1*1501 restricted CD4^+^ T cells could be less efficient in controlling EBV infection and stimulate autoimmunity in part via the resulting elevated levels of EBV transformed B cells. 

While there is evidence that EBV infection plays an important role in MS disease, a better understanding of the mechanisms by which EBV predisposes individuals to MS is needed. Therapeutic interventions that specifically target EBV infection could be a potential avenue for the future treatment of MS and help us understand whether and how EBV infection might contribute to the clinical course of MS.

## 5. Therapeutic Approaches Addressing EBV’s Contributions to MS

The effectiveness of interferon-beta (IFN-β), the first disease-modifying therapy that was able to reduce relapse rates in MS, was already associated with a reduction in EBV specific immune responses [74]. Furthermore, it has been shown that the treatment of MS with natalizumab, another disease-modifying therapy for active relapsing MS, led to the decline of anti-EBV gp350 levels in MS patients [75]. However, only the recent use of anti-CD20 therapies such as rituximab, ocrelizumab and ofatumumab in MS have provided remarkable effects and became the first therapies to slow down primary progressive disease [76,77,78]. How exactly B cell depletion mediates a beneficial effect in MS still remains unclear. One hypothesis is that the beneficial results stem from depletion of EBV infected B cell pools (Figure 1). While anti-CD20 therapies leave plasma cells and their produced antibodies such as OCBs untouched, they effectively deplete naïve and memory B cells, the primary site of persistent latent EBV infection, thereby eliminating the circulating EBV infected B cells and reducing the homing of autoreactive B and T cells to the CNS [79]. Indeed, it has been shown that after ocrelizumab treatment, EBV reactive T cell frequencies were decreased [80]. Additionally, some of those EBV transformed B cell reservoirs might exist in the CNS where B cells could act as antigen-presenting cells (APCs) for autoimmune T cells. To what extent anti-CD20 antibodies reach CNS resident EBV infected B cells is still unclear. However, the stimulation of autoreactive T cells may not exclusively happen in the CNS but rather in peripheral sites, such as the gut mucosa. Recent data suggest that an activation in the periphery of autoreactive T cells could trigger them to migrate and invade the CNS [81,82,83]. CD20-specific therapies could deplete such antigen-presenting, EBV transformed, memory B cells that promote MS. While anti-CD20 treatment rather argues for a decreasing EBV specific immune stimulation in MS by clearing EBV infected B cells, another therapeutic approach calls for an opposite intervention. 

A study using autologous T cells targeting the EBV infected B cell pool showed promising results that would argue for the strengthening of EBV specific T cell stimulation to combat MS (Figure 1). Ten secondary progressive MS patients received an autologous EBV specific T cell therapy, where patient-derived T cells were expanded in vitro with irradiated autologous peripheral blood mononuclear cells (PBMCs) that had been transduced with an adenoviral vector encoding EBNA1, LMP1 and LMP2A. This autologous EBV specific T cell product was then adoptively transferred back into the MS patients. This therapy resulted in clinical improvement with reduced disease activity that correlated with the EBV reactivity of the transferred T cells, and no serious adverse effects were reported [84,85]. While this EBV specific cell-based approach offers great possibilities with less off-target effects, strengthening EBV specific immune control to eliminate the infected B cell compartment is not without risk. Cross-reactivity and bystander damage from the boosted immune response could worsen MS through increased CNS inflammation [86]. Future research is needed with more clinical trials to assess the safety profile and duration of the effect. Therefore, it might be more beneficial to already strengthen the immune response towards EBV before primary infection occurs. 

EBV negative individuals have an extremely low risk to develop MS as EBV infection seems necessary for the disease development [21]. Thus, MS could potentially be prevented by a suitable vaccine against EBV. Especially EBV seronegative adolescents that are at higher risk of developing IM during primary infection and have a subsequent higher MS risk would benefit from such a vaccine [8]. Up to now there is no available vaccine that protects against EBV infection, but several different vaccinations are explored, including EBV derived virus-like particles, recombinant viral vectors and recombinant viral proteins [87,88,89,90,91,92]. It will be interesting to see if such a vaccine can prevent the development of IM or even MS in susceptible individuals. However, more research is still needed to better understand how EBV allows MS to develop and what is its role in the clinical course of MS. These findings could help to develop new treatments specifically targeting EBV, such as antiviral compounds, which could have major advantages in terms of specificity and off-target effects. 

## 6. Conclusions and Outlook

Strong epidemiological evidence has identified EBV infection as a prerequisite of MS development but the mechanisms behind this association remain unknown. Molecular mimicry and a deficient immune control of EBV in genetically MS susceptible individuals are two attractive and not mutually exclusive hypotheses for EBV’s role as one of the main environmental risk factors for MS development. Unfortunately, these two mechanisms would either require weakening cross-reactive immune responses or strengthening EBV specific immune control, respectively, requiring contrary therapeutic interventions. Therefore, more mechanistic insights into the association of the EBV with MS need to be gained.

Along these lines the contribution of EBV infection to the tertiary lymphoid structures in MS brains or autoimmune lymphocyte stimulation at other sites such as the gut [81,82,83] should be revisited and investigated. Accordingly, more investigations into B cell differentiation that allows CNS homing after EBV infection would be helpful, and primary CNS B cell lymphomas that are nearly uniformly EBV positive could provide interesting insights [93]. These and possibly CXCR3^+^ T-bet^+^ B cells should also be further explored for their T-cell-stimulating capacity, especially with respect to CNS autoreactivities [71]. In this respect LMP1 transgenic mouse B cells and EBV infected human B cells have been described to provide efficient antigen presenting functions for the stimulation of cytotoxic CD4^+^ T cells [94,95]. These investigations would address if a poorly controlled EBV infection in MS patients generates an antigen presenting cell compartment of B cells that drives autoimmunity in MS. If such an EBV infected B cell compartment could be identified, it might be targeted by antibody depletion using a more tailored approach than overall B cell depletion by anti-CD20 antibodies. Alternatively, EBV specific immune control could be strengthened either by EBV specific T cell transfer, as already therapeutically explored, or by vaccination.

For the molecular mimicry hypothesis, experimental systems that would allow to interrogate cross-reactive antibodies or T cells for their pathogenic functions would be required. Along these lines, EBV peptides have been recently used to modify EAE [46] but many EAE models are B-cell-independent. Therefore, ideally in vivo models of CNS disease dependent on B cells and susceptible to EBV infection should be developed to interrogate cross-reactivities between mostly EBNA1 or a few other EBV antigens and CNS autoantigens. The identification of CNS pathogenic immune specificities would then allow the development of tolerizing strategies for these EBV associated autoimmune responses, that could pave the way for future treatment options for MS. Along these lines, tolerization approaches with apoptotic cells or erythrocytes that are loaded with autoantigen-derived peptide epitopes continue to be therapeutically explored [96,97]. 

## Figures and Tables

**Figure 1 cells-11-02757-f001:**
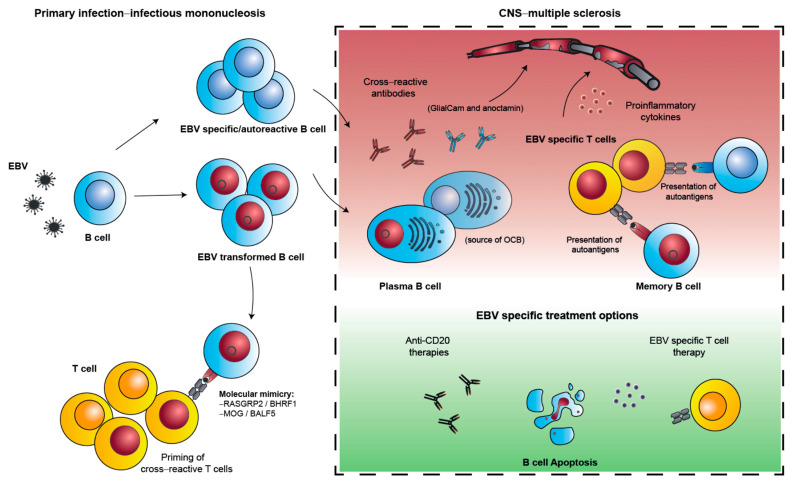
Potential mechanisms behind EBV’s association as a prerequisite of MS development. Strong immune activation during primary EBV infection, especially in genetically predisposed individuals could lead to insufficiently controlled EBV transformed B cells (represented as red nuclei). Those cells could allow the activation and expansion of autoreactive T cells (cross-reactivity depicted as red nuclei). Additionally, EBV infection could allow the proliferation and priming of noninfected, EBV and autoantigen specific B cells (depicted as blue nuclei). Through an unknown mechanism EBV transformed and EBV reactive B cells could home to the CNS, triggering the migration of other autoproliferative lymphocytes. In the CNS, plasma B cells producing cross-reactive antibodies against myelin antigens and memory B cells that act as APCs for the cross-reactive T cells trigger the production of proinflammatory cytokines. Together, this could start the immunopathology seen in MS as myelin damage. Treatments that either dampen the cross-reactive immune response (anti-CD20 treatment) or strengthen EBV specific immune control (EBV specific T cell therapy) might lead to the elimination EBV transformed B cells and ameliorate MS disease.

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
