# Peer review of "Altered Immune Response to the Epstein–Barr Virus as a Prerequisite for Multiple Sclerosis"

_cells, 2022, doi:10.3390/cells11172757_

Round 1

Reviewer 1 Report

In this review, the authors discussed potential role of EBV infection in the pathogenesis of multiple sclerosis. Some concerns are listed as below:

It is not clear for readers if EBV infection shares some potentially similar mechanisms regarding the development autoimmune diseases. Or MS disease specific?

I wonder if EBV infection is also a prerequisite for SLE or RA. This should be mentioned clearly.

Many healthy patients may also get EBV infection and they will not suffer from any autoimmune diseases. How do you explain this?

Some patients with MS will not get EBV infection in their life. How do you explain this?

How do you define 'EBV infected B cells'? Which methods can be used? This point should be introduced.

Apart from anti-CD20 therapies, what about the effects of other DMTs on EBV infected B cells since they can also decrease ARR?

How EBV infected B cells in the circulation cause CNS injury?Some may argue that the dose of Rituximab is low in the CNS (compared with the circulation). 

Molecular mimicry development during EBV infection should be shown using a figure.

Future perspectives should be provided by the authors in this review.

In Figure 1, the role of B cells (without EBV infection) in MS should be discussed.

Author Response

We thank both reviewers for their constructive comments that we have now incorporated into the revised manuscript version. The changes are out-lined below and marked by underlining in the revised manuscript.

In this review, the authors discussed potential role of EBV infection in the pathogenesis of multiple sclerosis. Some concerns are listed as below:

It is not clear for readers if EBV infection shares some potentially similar mechanisms regarding the development autoimmune diseases. Or MS disease specific? I wonder if EBV infection is also a prerequisite for SLE or RA. This should be mentioned clearly.

We have now clarified on page 2 that elevated EBV viral loads and immune responses might be rather a consequence of autoimmunity driven B cell differentiation in SLE and RA, while EBV infection seems to be a prerequisite to MS development: “In autoantibody driven diseases like RA and SLE elevated EBV viral loads might originate from lytic viral replication that arises secondary as a result from autoimmune B cell differentiation to plasma cells [1,2], but might nevertheless contribute to the inflammatory environment of these autoimmune diseases. In contrast, EBV’s mechanistic contribution to multiple sclerosis (MS) remains largely unknown and contrary to RA and SLE EBV dysregulation seems to be a prerequisite rather than a consequence of the autoimmune disease.”

Many healthy patients may also get EBV infection and they will not suffer from any autoimmune diseases. How do you explain this?

We have now specified on page 3 that in our opinion the current evidence is most consistent with EBV infection being a prerequisite for MS development that, however, requires additional genetic and environmental factors to lead to this autoimmune disease: “However, the majority of people infected with EBV will never develop MS. This indicates, that EBV infection is a prerequisite which allows the development of the autoimmune disease in genetically susceptible individuals that are exposed to additional environmental factors.”

Some patients with MS will not get EBV infection in their life. How do you explain this?

We consider this to be a rare event and even in the general population most seronegative young adults might carry EBV without the diagnostic antibody responses. We have now specified this on page 2: “While there are reports of EBV seronegative MS patients, those cases are extremely rare. It is even considered that the majority of these EBV seronegative individuals are misdiagnosed, since serological tests against multiple antigens are recommended to accurately define EBV status [3].”

How do you define 'EBV infected B cells'? Which methods can be used? This point should be introduced.

We have now modified the introduction on page 1 to specify that while all EBV proteins can be down-regulated during infected B cell differentiation, the EBV encoded small non-translated RNAs (EBERs) remain expressed and their detection remains a reliable method to detect EBV infected B cells: “Once the virus has gained access to the memory B cell pool, EBV can persist without the expression of any viral proteins, the so called latency 0. However, EBV–encoded small non-coding RNAs (EBERs) are still expressed and can be detected by in situ hybridization [4].”

Apart from anti-CD20 therapies, what about the effects of other DMTs on EBV infected B cells since they can also decrease ARR?

We have now added a brief discussion on interferon-beta and alpha4-integrin blocking therapy (Natalizumab). On page 8 we state: “The effectiveness of interferon-beta (IFN-β), the first disease-modifying therapy that was able to reduce relapse rates in MS, was already associated with a reduction in EBV-specific immune responses [5]. Furthermore, it has been shown that treatment of MS with Natalizumab, another disease modifying therapy for active relapsing MS, led to the decline of anti-EBV gp350 levels in MS patients [6].”

How EBV infected B cells in the circulation cause CNS injury? Some may argue that the dose of Rituximab is low in the CNS (compared with the circulation).

On page 8 of the revised manuscript we now address this concern of the reviewer: “To what extent anti-CD20 antibodies reach CNS resident EBV infected B cells is still unclear. However, stimulation of autoreactive T cells may not exclusively happen in the CNS but rather in peripheral sites, such as the gut mucosa. Recent data suggest that activation in the periphery of autoreactive T cells could trigger them to migrate and invade the CNS [7-9].”

Molecular mimicry development during EBV infection should be shown using a figure.

We have now modified figure 1 to include molecular mimicry for both antibody and T cell recognition.

Future perspectives should be provided by the authors in this review.

We have now extended the outlook on page 10 to include insights into the mechanistic connection between EBV infection and MS that can be gained from experimental MS therapies in the future: “If such a EBV infected B cell compartment could be identified it might be targeted by antibody depletion using a more tailored approach than overall B cell depletion by anti-CD20 antibodies. Alternatively, EBV specific immune control could be strength-ened either by EBV specific T cell transfer, as already therapeutically explored, or by vaccination.” and “Along these lines tolerization approaches with apoptotic cells or erythrocytes that are loaded with autoantigen derived peptide epitopes continue to be therapeutically ex-plored [10,11].”

In Figure 1, the role of B cells (without EBV infection) in MS should be discussed.

Non-infected B cells and their possible role in autoantibody production and autoantigen presentation have now been added to figure 1.

References

  1. Lünemann, J.D.; Frey, O.; Eidner, T.; Baier, M.; Roberts, S.; Sashihara, J.; Volkmer, R.; Cohen, J.D.; Hein, G.; Kamradt, T., et al. Increased frequency of EBV specific effector memory CD8+ T cells is associated with higher viral load in rheumatoid arthritis. J Immunol 2008, 181, 991-1000.
  2. Laichalk, L.L.; Thorley-Lawson, D.A. Terminal differentiation into plasma cells initiates the replicative cycle of Epstein-Barr virus in vivo. J Virol 2005, 79, 1296-1307.
  3. Endriz, J.; Ho, P.P.; Steinman, L. Time correlation between mononucleosis and initial symptoms of MS. Neurol Neuroimmunol Neuroinflamm 2017, 4, e308.
  4. Weiss, L.M.; Chen, Y.Y. EBER in situ hybridization for Epstein-Barr virus. Methods Mol Biol 2013, 999, 223-230.
  5. Comabella, M.; Kakalacheva, K.; Rio, J.; Münz, C.; Montalban, X.; Lunemann, J.D. EBV-specific immune responses in patients with multiple sclerosis responding to IFNbeta therapy. Mult Scler 2012, 18, 605-609.
  6. Persson Berg, L.; Eriksson, M.; Longhi, S.; Kockum, I.; Warnke, C.; Thomsson, E.; Backstrom, M.; Olsson, T.; Fogdell-Hahn, A.; Bergstrom, T. Serum IgG levels to Epstein-Barr and measles viruses in patients with multiple sclerosis during Natalizumab and interferon beta treatment. BMJ Neurol Open 2022, 4, e000271.
  7. Berer, K.; Gerdes, L.A.; Cekanaviciute, E.; Jia, X.; Xiao, L.; Xia, Z.; Liu, C.; Klotz, L.; Stauffer, U.; Baranzini, S.E., et al. Gut microbiota from multiple sclerosis patients enables spontaneous autoimmune encephalomyelitis in mice. Proc Natl Acad Sci U S A 2017, 114, 10719-10724.
  8. Berer, K.; Mues, M.; Koutrolos, M.; Rasbi, Z.A.; Boziki, M.; Johner, C.; Wekerle, H.; Krishnamoorthy, G. Commensal microbiota and myelin autoantigen cooperate to trigger autoimmune demyelination. Nature 2011, 479, 538-541.
  9. Wekerle, H.; Berer, K.; Krishnamoorthy, G. Remote control-triggering of brain autoimmune disease in the gut. Curr Opin Immunol 2013, 25, 683-689.
  10. McRae, B.L.; Vanderlugt, C.L.; Dal Canto, M.C.; Miller, S.D. Functional evidence for epitope spreading in the relapsing pathology of experimental autoimmune encephalomyelitis. J Exp Med 1995, 182, 75-85.
  11. Lutterotti, A.; Yousef, S.; Sputtek, A.; Sturner, K.H.; Stellmann, J.P.; Breiden, P.; Reinhardt, S.; Schulze, C.; Bester, M.; Heesen, C., et al. Antigen-specific tolerance by autologous myelin peptide-coupled cells: A phase 1 trial in multiple sclerosis. Sci Transl Med 2013, 5, 188ra175.
  12. Ghareghani, M.; Reiter, R.J.; Zibara, K.; Farhadi, N. Latitude, vitamin D, melatonin, and gut microbiota act in concert to initiate multiple sclerosis: A new mechanistic pathway. Front Immunol 2018, 9, 2484.
  13. Munger, K.L.; Zhang, S.M.; O'Reilly, E.; Hernan, M.A.; Olek, M.J.; Willett, W.C.; Ascherio, A. Vitamin D intake and incidence of multiple sclerosis. Neurology 2004, 62, 60-65.
  14. Smolders, J.; Thewissen, M.; Peelen, E.; Menheere, P.; Tervaert, J.W.; Damoiseaux, J.; Hupperts, R. Vitamin D status is positively correlated with regulatory T cell function in patients with multiple sclerosis. PLoS One 2009, 4, e6635.
  15. Wingate, P.J.; McAulay, K.A.; Anthony, I.C.; Crawford, D.H. Regulatory T cell activity in primary and persistent Epstein-Barr virus infection. J Med Virol 2009, 81, 870-877.

Reviewer 2 Report

This is a well-written review paper that discusses the potential mechanisms by which Epstein Barr virus (EBV) infection predisposes to the development of multiple sclerosis (MS). The paper covers the proposed role of EBV in several autoimmune diseases and its now well-documented role in MS, the timing of EBV infection relative to MS development, issues related to molecular mimicry and altered immune responses to EBV in MS patients, and potential therapeutic approaches based on EBV’s involvement in MS. It is argued that mechanisms will need to be further explored and that developing a vaccine to EBV should have high priority. A nice illustration is included.

Minor comments:

1.     Can you comment on the potential role of vitamin D/geographic location as it relates to EBV infection and MS?

2.     Typos: 

Line 31: replace “cells” by “cell
Line 34: replace “EBVs” by “EBV’s”

Line 54: replace “EBVs” by “EBV’s”

Line 55: replace “Patient” by “Patients”

Line 177: replace “compartimentalized” by “compartmentalized”

Line 236: replace “10” by “Ten”

Line 253: replace “adolescent” by “adolescents”

Line 269: replace “EBVs” by “EBV’s”

Lines 276-283: “Along these lines” is used 3 times, which is a bit repetitive

Line 283: replace “mouse” by “mice”

Author Response

We thank both reviewers for their constructive comments that we have now incorporated into the revised manuscript version. The changes are out-lined below and marked by underlining in the revised manuscript.

This is a well-written review paper that discusses the potential mechanisms by which Epstein Barr virus (EBV) infection predisposes to the development of multiple sclerosis (MS). The paper covers the proposed role of EBV in several autoimmune diseases and its now well-documented role in MS, the timing of EBV infection relative to MS development, issues related to molecular mimicry and altered immune responses to EBV in MS patients, and potential therapeutic approaches based on EBV’s involvement in MS. It is argued that mechanisms will need to be further explored and that developing a vaccine to EBV should have high priority. A nice illustration is included.

Minor comments:

  1. Can you comment on the potential role of vitamin D/geographic location as it relates to EBV infection and MS?

On page 3 of the revised manuscript we have now added brief discussions on the association of low vitamin D with MS and a possible role for low vitamin D in IM development: “One reason for this could be the limited sunlight exposure and hence lower vitamin D levels in countries with high latitude [12]. Indeed, vitamin D intake was shown to decrease the risk for MS development [13].” and “There are indications that the low vitamin D level could be partially responsible for this geographical distribution of IM occurrence. Serum levels of 25- hydroxyvitamin D were shown to positively correlate with the ability of regulatory T cells to suppress T cell proliferation. This ability plays an important part during primary EBV infection, in controlling the EBV specific T cell response, thereby preventing immunopathologies as seen in IM [14,15].”

  1. Typos:

Line 31: replace “cells” by “cell

Line 34: replace “EBVs” by “EBV’s”

Line 54: replace “EBVs” by “EBV’s”

Line 55: replace “Patient” by “Patients”

Line 177: replace “compartimentalized” by “compartmentalized”

Line 236: replace “10” by “Ten”

Line 253: replace “adolescent” by “adolescents”

Line 269: replace “EBVs” by “EBV’s”

Lines 276-283: “Along these lines” is used 3 times, which is a bit repetitive

Line 283: replace “mouse” by “mice”

These typos and redundancies have now been corrected.

References

  1. Lünemann, J.D.; Frey, O.; Eidner, T.; Baier, M.; Roberts, S.; Sashihara, J.; Volkmer, R.; Cohen, J.D.; Hein, G.; Kamradt, T., et al. Increased frequency of EBV specific effector memory CD8+ T cells is associated with higher viral load in rheumatoid arthritis. J Immunol 2008, 181, 991-1000.
  2. Laichalk, L.L.; Thorley-Lawson, D.A. Terminal differentiation into plasma cells initiates the replicative cycle of Epstein-Barr virus in vivo. J Virol 2005, 79, 1296-1307.
  3. Endriz, J.; Ho, P.P.; Steinman, L. Time correlation between mononucleosis and initial symptoms of MS. Neurol Neuroimmunol Neuroinflamm 2017, 4, e308.
  4. Weiss, L.M.; Chen, Y.Y. EBER in situ hybridization for Epstein-Barr virus. Methods Mol Biol 2013, 999, 223-230.
  5. Comabella, M.; Kakalacheva, K.; Rio, J.; Münz, C.; Montalban, X.; Lunemann, J.D. EBV-specific immune responses in patients with multiple sclerosis responding to IFNbeta therapy. Mult Scler 2012, 18, 605-609.
  6. Persson Berg, L.; Eriksson, M.; Longhi, S.; Kockum, I.; Warnke, C.; Thomsson, E.; Backstrom, M.; Olsson, T.; Fogdell-Hahn, A.; Bergstrom, T. Serum IgG levels to Epstein-Barr and measles viruses in patients with multiple sclerosis during Natalizumab and interferon beta treatment. BMJ Neurol Open 2022, 4, e000271.
  7. Berer, K.; Gerdes, L.A.; Cekanaviciute, E.; Jia, X.; Xiao, L.; Xia, Z.; Liu, C.; Klotz, L.; Stauffer, U.; Baranzini, S.E., et al. Gut microbiota from multiple sclerosis patients enables spontaneous autoimmune encephalomyelitis in mice. Proc Natl Acad Sci U S A 2017, 114, 10719-10724.
  8. Berer, K.; Mues, M.; Koutrolos, M.; Rasbi, Z.A.; Boziki, M.; Johner, C.; Wekerle, H.; Krishnamoorthy, G. Commensal microbiota and myelin autoantigen cooperate to trigger autoimmune demyelination. Nature 2011, 479, 538-541.
  9. Wekerle, H.; Berer, K.; Krishnamoorthy, G. Remote control-triggering of brain autoimmune disease in the gut. Curr Opin Immunol 2013, 25, 683-689.
  10. McRae, B.L.; Vanderlugt, C.L.; Dal Canto, M.C.; Miller, S.D. Functional evidence for epitope spreading in the relapsing pathology of experimental autoimmune encephalomyelitis. J Exp Med 1995, 182, 75-85.
  11. Lutterotti, A.; Yousef, S.; Sputtek, A.; Sturner, K.H.; Stellmann, J.P.; Breiden, P.; Reinhardt, S.; Schulze, C.; Bester, M.; Heesen, C., et al. Antigen-specific tolerance by autologous myelin peptide-coupled cells: A phase 1 trial in multiple sclerosis. Sci Transl Med 2013, 5, 188ra175.
  12. Ghareghani, M.; Reiter, R.J.; Zibara, K.; Farhadi, N. Latitude, vitamin D, melatonin, and gut microbiota act in concert to initiate multiple sclerosis: A new mechanistic pathway. Front Immunol 2018, 9, 2484.
  13. Munger, K.L.; Zhang, S.M.; O'Reilly, E.; Hernan, M.A.; Olek, M.J.; Willett, W.C.; Ascherio, A. Vitamin D intake and incidence of multiple sclerosis. Neurology 2004, 62, 60-65.
  14. Smolders, J.; Thewissen, M.; Peelen, E.; Menheere, P.; Tervaert, J.W.; Damoiseaux, J.; Hupperts, R. Vitamin D status is positively correlated with regulatory T cell function in patients with multiple sclerosis. PLoS One 2009, 4, e6635.
  15. Wingate, P.J.; McAulay, K.A.; Anthony, I.C.; Crawford, D.H. Regulatory T cell activity in primary and persistent Epstein-Barr virus infection. J Med Virol 2009, 81, 870-877.

Round 2

Reviewer 1 Report

The authors have addressed my concerns.

The authors said that the majority of these EBV seronegative individuals are misdiagnosed in the revised manuscript. This statement should be considered carefully.

Author Response

We thank the second reviewer for his/her additional comment which we have now addressed in the revised manuscript version. The changes are out-lined below and marked by underlining in the revised manuscript.

Reviewer #1

The authors have addressed my concerns.

The authors said that the majority of these EBV seronegative individuals are misdiagnosed in the revised manuscript. This statement should be considered carefully.

Thank you for this comment. This is indeed a not too well investigated but very interesting area of EBV immunobiology. We have now added additional evidence that EBV specific T cell responses and viral loads can be detected in the majority of EBV seronegative adults. Since these were, however, not MS patients, we also toned down our statement on page 2 of the revised manuscript: “While there are reports of EBV seronegative MS patients, those cases are extremely rare. It is even considered that the majority of EBV seronegative adults may be misdiagnosed as uninfected, since serological tests against multiple antigens are recommended to accurately define EBV status, and in the majority of EBV seronegative adults but not seronegative children viral loads and EBV specific T cell responses could previously be detected [1,2]. “

References

  1. Savoldo, B.; Cubbage, M.L.; Durett, A.G.; Goss, J.; Huls, M.H.; Liu, Z.; Teresita, L.; Gee, A.P.; Ling, P.D.; Brenner, M.K., et al. Generation of EBV-specific CD4+ cytotoxic T cells from virus naive individuals. J Immunol 2002, 168, 909-918.
  2. Endriz, J.; Ho, P.P.; Steinman, L. Time correlation between mononucleosis and initial symptoms of MS. Neurol Neuroimmunol Neuroinflamm 2017, 4, e308.